# Inclusion Biogenesis, Methods of Isolation and Clinical Application of Human Cellular Exosomes

**DOI:** 10.3390/jcm9020436

**Published:** 2020-02-06

**Authors:** Max Tschuschke, Ievgeniia Kocherova, Artur Bryja, Paul Mozdziak, Ana Angelova Volponi, Krzysztof Janowicz, Rafał Sibiak, Hanna Piotrowska-Kempisty, Dariusz Iżycki, Dorota Bukowska, Paweł Antosik, Jamil A. Shibli, Marta Dyszkiewicz-Konwińska, Bartosz Kempisty

**Affiliations:** 1Department of Anatomy, Poznan University of Medical Sciences, 60-781 Poznań, Poland; maxtsch@wp.pl (M.T.); ikocherova@ump.edu.pl (I.K.); abryja@ump.edu.pl (A.B.); krzysztof.janowicz.16@abdn.ac.uk (K.J.); m.dyszkiewicz@ump.edu.pl (M.D.-K.); 2Physiology Graduate Program, North Carolina State University, Raleigh, NC 27695, USA; pemozdzi@ncsu.edu; 3Centre for Craniofacial and Regenerative Biology, Faculty for Dentistry, Oral and Craniofacial Sciences, King’s College University of London, London SE1 9RT, UK; ana.angelova@kcl.ac.uk; 4The School of Medicine, Medical Sciences and Nutrition, University of Aberdeen, Aberdeen AB25 2ZD, UK; 5Division of Reproduction, Department of Obstetrics, Gynecology, and Gynecologic Oncology, Poznan University of Medical Sciences, 60-535 Poznan, Poland; 75094@student.ump.edu.pl; 6Department of Toxicology, Poznan University of Medical Sciences, 61-131 Poznań, Poland; hpiotrow@ump.edu.pl; 7Department of Cancer Immunology, Poznan University of Medical Sciences, 61-866 Poznań, Poland; dmizy@ump.edu.pl; 8Department of Diagnostics and Clinical Sciences, Nicolaus Copernicus University in Torun, 87-100 Toruń, Poland; dbukowska@umk.pl; 9Department of Veterinary Surgery, Nicolaus Copernicus University in Torun, 87-100 Toruń, Poland; pantosik@umk.pl; 10Department of Periodontology and Oral Implantology, Dental Research Division, University of Guarulhos, Guarulhos 07030-010, Brazil; jashibli@yahoo.com; 11Department of Biomaterials and Experimental Dentistry, Poznan University of Medical Sciences, 61-701 Poznan, Poland; 12Department of Histology and Embryology, Poznan University of Medical Sciences, 60-781 Poznań, Poland; 13Department of Obstetrics and Gynaecology, University Hospital and Masaryk University, 601 77 Brno, Czech Republic; 14Institute of Veterinary Medicine, Nicolaus Copernicus University in Toruń, 87-100 Toruń, Poland

**Keywords:** exosome, neurodegenerative disease, cancer, biomarker, clinical application

## Abstract

Exosomes are a heterogenous subpopulation of extracellular vesicles 30–150 nm in range and of endosome-derived origin. We explored the exosome formation through different systems, including the endosomal sorting complex required for transport (ESCRT) and ESCRT-independent system, looking at the mechanisms of release. Different isolation techniques and specificities of exosomes from different tissues and cells are also discussed. Despite more than 30 years of research that followed their definition and indicated their important role in cellular physiology, the exosome biology is still in its infancy with rapidly growing interest. The reasons for the rapid increase in interest with respect to exosome biology is because they provide means of intercellular communication and transmission of macromolecules between cells, with a potential role in the development of diseases. Moreover, they have been investigated as prognostic biomarkers, with a potential for further development as diagnostic tools for neurodegenerative diseases and cancer. The interest grows further with the fact that exosomes were reported as useful vectors for drugs.

## 1. Introduction

In order to maintain homeostasis, cells continuously interact with their environment through the secretion of different types of extracellular vesicles. Extracellular vesicles (EVs), comprising of a heterogenous group of membrane-derived vesicles of varying origin, size, and features, have a crucial role in cellular exchange. Despite the fact that the term has been broadly used for various forms of EVs [1], basic criteria for their definition have been determined [2]. The main division and separation of nanovesicles are based on the process of biogenesis, size of the vesicles, and cargos [3]. The largest are apoptotic bodies produced by cells during apoptosis, 1–5 µm in diameter, and generated by budding directly from the plasma membrane (PM), followed by release into extracellular space [4,5]. Microvesicles (MV) are 150–1000 nm vesicles that have a similar method of formation as the apoptotic bodies [6]. The smallest and most recently discovered subpopulation of nanovesicles are exosomes, cellular mediators with a diameter of 30 to 150 nm [7]. Exosomes are formed differently than microvesicles and apoptotic bodies (Figure 1), through the invagination of endosomal membrane, resulting in multivesicular body (MVBs) formation, which later fuses with PM and releases exosomes into the extracellular space [8]. Even though characteristics of microvesicles, apoptotic bodies and exosomes are well understood, the size ranges are only rough estimates. Exosomes are produced by a majority of mammalian cells, such as: B lymphocytes, cytotoxic cells, platelets, oligodendrocytes, dendritic cells, mast cells, adipocytes, neurons, glial cells, endothelial cells and epithelial cells [5,9]. Exosomes’ release takes place both in physiological and morbid conditions, with these nanovesicles present in various body fluids [10]. For the first time, exosomes were observed in 1983, by two independent groups of researchers [11,12]. They described the externalization of transferrin receptors during the maturation of a sheep’s reticulocytes via small vesicles of 50 nm in size. The term “exosome”, defining those structures, was used four years later [13]. At the beginning, exosomes were considered only as cellular disposal of obsolete proteins and other molecules [14]. However, subsequent studies confirmed their functions in continuous intercellular communication. In 1996, Raposo et al. reported their involvement in antigen presentation and adaptive immune response. It was shown that proteins bound to major histocompatibility complex (MHC) class II dimers placed on exosomes, which were produced and secreted by Epstein-Barr-virus-transformed B lymphocytes, induced stimulation of specific T cells [15]. In 1998, another group of researchers described exosomes’ secretion by dendritic cells promoting antitumor response [16]. Since then, numerous publications described the important role of exosomes in cell-to-cell communication, carrying various molecular cargo [17]. The current version of ExoCarta online database hosts 41,860 proteins, >7540 RNA, and 1116 lipids that can be found in exosomes [18]. Other exosomes dedicated databases with less entries include Exosome RNA, Vesiclepedia, Urinary Exosome Protein Database, exoRBase, and EVpedia. This variety of molecules proves a significant role of nanovesicles in numerous physiological processes, such as lactation, cell proliferation and immune response [19,20,21], but also in pathological states like cardiovascular diseases, neurodegenerative process, cancer development and progression, inflammation, or even asthma.

## 2. Cellular Origins and Chemical Properties of Exosomes

Exosome biogenesis is inseparably connected with the endocytic pathway (Figure 2), such that invagination of plasma membrane during endocytosis results in early endosome formation (EE). Maturation of EEs into late endosomes (LE) occurs via inward budding of early endosome membranes. Within lumen LE develops as the multivesicular body (MVB) containing intraluminal vesicles (ILVs) [22]. Most of the time, MVBs are directed to lysosomes containing hydrolase, resulting in degradation of their cargo [23]. Otherwise, MVBs migrate to the cell surface to fuse with PM and release ILVs into the extracellular space, that in turn become an exosome upon cellular exit [24]. Transport of MVBs is directed through accessory proteins: tumor susceptibility gene 101 protein (TSG101), programmed cell death 6-interacting protein Alix, heat shock cognate protein 70 (HSC70), heat shock protein 90β (HSP90β), cluster of differentiation proteins 9 (CD9), CD81, CD63, and involves either the presence of ESCRT protein family, known as exosomal marker family proteins crucial in the ESCRT-dependent formation or alternatively sphingomyelinase enzyme in the ESCRT-independent formation [25]. Tetraspanin enriched microdomeins (TEM) assisted by CD81 play major role in sorting of bioactive proteins, genetic materials and lipids into the exosomes [26]. Zhang et al. recently described the establishment of exosome based intracellular communication being possible due to microRNAs, which are highly abundant as exosomal cargo mainly demonstrate their function in human plasma derived exosomal species. 

### 2.1. ESCRT-Dependent Formation

Numerous pathways and molecules are involved in formation of MVBs and ILVs. The endosomal sorting complex required for transport (ESCRT) is the most known ubiquitin-dependent mechanism responsible for sorting ubiquitinated proteins into ILVs [27]. This mechanism was presented in Figure 3. ESCRT consists of four complexes numbered in order of their action: ESCRT-0 (previously called vacuolar protein sorting-associated protein 27/heat shock element 1 complex VPS27/HSE1), ESCRT-I, ESCRT-II, and ESCRT-III. These complexes cooperate with specific molecules, such as: VPS4 proteins (VPS4A, VPS4B, lyst-interacting protein 5 (LIP5)) and Bro1 complexes (Alix, his-domain protein-tyrosine phosphatase (HDPTP), BRO1 domain and CAAX motif containing protein (BROX)) [22]. ESCRT-0 is activated by phosphatidylinositol 3-phosphate PI(3)P and ubiquitinated molecules present on the outside of endosomal membrane. The whole process initiates by recognising and engaging ubiquitinated transmembrane proteins, promoting their concentration on the late endosomal membrane. ESCRT-0 also recruits ESCRT-I through the interaction between hepatocyte growth factor regulated tyrosine kinase substrate prosaposin (HRS PSAP) domains and ESCRT-I subunit TSG101 [28]. ESCRT-I complex is essential for sorting cargo in the MVB and deforming the membrane, resulting in bud formation. ESCRT-II also participates in cargo sorting, additionally regulating ESCRT-III complex formation [29]. Progida et al. suggested ESCRT-II interaction with RILP protein, which also binds with dynein–dynactin motor complex, involved in endosome motility [30]. ESCRT-III is responsible for the sorting and concentration of MVB cargo, as well as driving vesicle scission. It also participates in ESCRT recycling via recruitment of the AAA-type VPS4 ATP-ase [31].

### 2.2. ESCRT-Independent Pathways

The ESCRT pathway is considered to be the most important mechanism of exosome formation. However, MBV and ILV formation also occur in a ubiquitin-independent way. Heparan sulphate proteoglycans promote exosome biogenesis through syntenin, a cytosolic adaptive protein. Syntenin binds syndecan with Alix, which interacts with several ESCRT (TSG101 and charged multivesicular body protein 4 (CHMP4)) proteins. It serves as an intermediate between ESCRT-I and ESCRT-III, and is involved in the budding and scission processes [7]. Moreover, recent articles indicate the presence of an ESCRT-independent pathway for exosome formation (Figure 4). ESCRT-independent formation was initially described in oligodendroglial cells that secreted exosomes containing proteolipid protein (PLP) [32]. The secretion of exosomes happened only after depletion of neutral sphingomyelinases (nSMase), enzymes hydrolysing sphingomyelin to ceramide, being indicative as a crucial role of ceramide in PLP sorting into ILVs. Interestingly, exosome secretion was not decreased, despite ESCRT inhibition. Recently, four-transmembrane domain proteins belonging to the tetraspanin family have been considered to be implicated in other pathways of cargo selection and exosome formation, in neither ESCRT-dependent nor ceramide-dependent manner [33]. Human melanoma cells secrete exosomes containing melanosome proteins, following a CD63-dependent mechanism [34]. In human embryonic kidney 293 cells (HEK293), the expression of CD82 and CD9 promote the discharge of β-catenin through exosomes, whereas a study employing rat pancreatic adenocarcinoma cells describes a role for Tspan8 in the recruitment of particular mRNA and transmembrane proteins into exosomes [35,36]. In 2013, Perez-Hernandez et al. described Tetraspanin-enriched microdomains (TEM) full of CD81 particles, which is considered to be another ESCRT-independent manner of protein sorting into ILVs [37]. There are far more molecules and cellular structures creating various pathways of exosome formation, such as lipid raft domains [38], flotllin-2 [39], phospholipase D2 (PLD2) and GTPase ADP ribosylation factor 6 (ARF6) [40,41], chaperone HSC70 [42] and membrane protein of lysosomes and late endosomes called lipopolysaccharide induced TNF factor (SIMPLE) [43]. In conclusion, there are many pathways of exosome formation and it is still unclear whether the sorting and sequestering of particular molecules involve different mechanisms, and consequently proteins, or if there are various MVB subpopulations within singular cells.

### 2.3. Secretion

Just as multiple pathways occur during MVB biogenesis, numerous mechanisms and particles are considered to be involved in exosome secretion. After formation, MVB can either fuse with a lysosome to degrade their cargo or fuse with the plasma membrane, resulting in exosome release. Although different MVB fates are known, the mechanism distinguishing both paths remains not fully understood. A recent study by Villarroya-Beltri suggests that ISGylation, a posttranslational ubiquitin-like modification of TSG101 (one of the ESCRT-I complex components) induces its aggregation and degradation, promoting exosome release by stimulation of fusion of MVB with lysosome. This results in increased exosome release indicating posttranslational modifications of the cargo proteins as a regulatory mechanism of MVB fate determination. Continuous transport of MVB to plasma membrane is performed via their interaction with actin, cortactin, microtubule skeleton and RAB proteins, along with their effectors [44]. Hoshino et al. suggested the role of actin cytoskeleton in exosome release in cancer cells. Actin-rich invadopodia seem to have remarkable influence on the secretion of exosomes [45]. MVB fusion with plasma membrane is facilitated and controlled by the largest family of more than 60 small GTPases-RAB proteins, which participate in all of the processes concerning vesicle transport within cells. The details of their action remain unknown, although involvement of specific proteins, namely RAB2B, RAB4, RAB5A, RAB7, RAB9A, RAB11, RAB27A, RAB27B, RAB35, is already confirmed in endosome motility and exosome secretion. Generally, release of exosomes requires various RAB proteins. After MVB docking with plasma membrane, SNARE (soluble NSF-attachment protein receptor) complexes facilitate the fusion of plasma membrane and MVB employing the SNAP protein. Until now, two proteins of the SNARE family facilitate exosome release: VAMP7 is crucial for the secretion of acetylcholinesterase-containing vesicles from K562 erythroleukemia cells, while YKT6 is necessary for WNT3A release from HEK293 cells. Apart from RAB and SNARE family, there are other effectors considered to be involved in exosome secretion, such as diacyl glycerol kinase α (DGKα), V0 subunit of vacuolar ATPase (V0-ATPase) and small GTPases of the Rho/Rac/cdc42 family [22,46,47].

### 2.4. Cargo 

Exosomes are 30–150 nm in size, double-layered vesicles with density fluctuation from 1.10 to 1.20 g/mL [48]. Exosomes have cup-shaped or saucer-like morphology, observable under transmission electron microscopy [49]. The cup-shape of EVs is observed after negative staining or other non-cryo EM processing, i.e., it has been interpreted as an artefact of this processing as the cup-shape is not visible by cryo-EM [50]. Regarding biochemical properties, exosomes represent a heterogenous family of vesicles marked by different compositions and carrying diverse cargo [51]. The diversity of exosome secreting cells results in a different protein composition of each exosome subpopulation, although, due to endosomal origins, analogous molecules are commonly found in a majority of these subpopulations [25]. The common group comprises proteins crucial for MVB biogenesis (Alix, TSG101, clathrin), molecules involved in fusion and exosome release (RAB and other families of small GTPases, flotollins, annexins, ARF6), adhesion proteins (integrins), tetraspanins (CD9, CD63, CD81), heat-shock proteins (HSP70, HSP90), cytoskeletal proteins (actin, tubulin) and metabolic enzymes (e.g., aldolase 1, GAPDH, PKM2). The presence of other proteins, such as MHC II depends on specific exosome releasing cells [52].

Even though the EVs always have a lipid encapsulation, lipidomics is not the most often used method in EV analysis, and there are few studies regarding proteomic analysis, highlighting as with proteins, different cells release exosomes of diverse lipid composition [53]. The exosome membrane does not exactly reflect plasma membrane of the maternal cell (e.g., enrichment of exosome membrane in sphingomyelin and glycosphingolipids, as well as lower amount of phosphatidylinositol in comparison to releasing cell) [28]. However, similarly to PM, the exosome membrane is double-layered with an asymmetric distribution of particular lipid classes in inner and outer portions [54]. Exosomes are enriched in sphingolipids, especially sphingomyelin, cholesterol, phosphatidylcholine, phosphatidylserine (PS), phosphatidylethanolamine, ceramide, and glycerophospholipids [55]. Curiously, there is an absence of lysobisphosphatidic acid (LBPA), which is crucial for ILV formation and is found inside those structures. As a plasma membrane, the exosome membrane contains lipid rafts, detergent-resistant domains containing specific components (e.g., glycolipids, Src tyrosine kinases or glycosylphosphatidylinositol (GPI-anchored proteins) [56]. Lipid rafts are involved in exosome formation and the secretion of specific molecules into the extracellular space [38].

Apart from proteins and lipids, exosomes carry a significant amount of RNA cargo, including mRNA, miRNA, and other non-coding RNAs, such as lncRNA [57]. In 2013, Valadi et al. described 1300 mRNA and 120 miRNA particles that are transported to target cells via exosomes, which may regulate their gene expression and protein translation [46]. This horizontal transport of functional RNA between cells has been observed in vitro as well as in vivo [58,59,60,61,62,63]. Exosomal RNA differs from maternal cell RNA content, which proves the existence of specific mechanisms and proteins controlling RNA sorting into exosomes [64]. Even though the nature of this process remains unknown, RNA binding proteins (RBP), such as Mex-3 RNA Binding Family Member C (MEX3C) and heterogeneous nuclear ribonucleoprotein A2/B1 (hnRNPA2B1) are some of the molecules suggested to be involved in the sorting of miRNA [65]. Next to RNA, several studies have described the existence of exosomal DNA (exoDNA) which, in contrary to exoRNA, probably undergoes a random sorting process. Thereby, exoDNA reflects the complete genomic DNA of the parental cell [66]. Other studies confirm these results suggesting that exosomes carry fetal cfDNA, that can be used as a biomarker for pregnancy complications [67]. On the contrary, study by Jeppesen et al. contradicts feature of exosome as active vehicles for DNA release, suggesting that DNA is more likely released through endosomal mechanisms and autophagy [68]. 

## 3. Methods of Isolation and Specificity of Exosomes Isolated from Selected Tissue and Cell Types

Extracellular vesicles, including exosomes, have been isolated from fluids, e.g., plasma, saliva, human breast milk, semen, amniotic fluid, cerebrospinal fluid, bronchoalveolar lavage, bile, urine, synovial fluid, aqueous humour, tear fluid, nasal secretions, and pleural effusions [2,69]. However, samples derived from biofluids contain an exosome mix of different cellular origins. To analyse exosomes from particular tissues or cells, it is best to collect conditioned media from cultured cells [70]. Currently, there is intensive research going on, looking at different aspects of the exosomes and focusing on their isolation. Various methods of isolation, purification, and further characterization are being developed concurrently, however the isolation methods do not only isolate exosomes or nanovesicles but also other precipitates and contaminants. Several methods of exosome isolation have been proposed and developed: ultracentrifugation, ultrafiltration, size exclusion chromatography (SEC), polymer precipitation, immunoaffinity chromatography and techniques based on microfluids. Each method and approach have advantages, as well as disadvantages, and is used depending on the size of exosomes and their origins.

### 3.1. Ultracentrifugation

Differential ultracentrifugation is the most commonly used technique of exosome isolation from biofluids and cell cultures [71]. It consists of three centrifugation steps with increasing centrifugal forces. First, low-speed centrifugation (300× *g*) is performed to remove cells and large cell debris from the cell culture fluid. The second round of centrifugation (10,000–20,000× *g*) is applied to remove large cellular debris, organelles and MVs. The last round of centrifugation is performed at highest speed (100,000–150,000× *g*) in order to separate exosomes from the supernatant. To produce exosome preparations of higher purity, a sucrose or iodoxinol density-gradient medium used to separate exosomes from other nonvesicles according to molar concentration and thus the density of particular phases. Ultracentrifugation is quite expensive, time-consuming, and a large amount of untreated samples is used, with the possibility of damaging the exosomes during the procedure [21,72].

### 3.2. Size-Based Isolation of Exosomes

This term comprises ultrafiltration and size exclusion chromatography (SEC). These methods are based on a passage through physical barriers dependent on the size of particles. Ultrafiltration uses nanomembranes or membranes with different cut-off molecular weights (MWCO) [73]. SEC employs columns containing heterogenous pours. These methods do not require special equipment and do not pose a danger of damaging exosomes during the procedure. When it comes to disadvantages, the SEC method is relatively time-consuming. Additionally, molecules of the same size range cannot be separated from exosomes. In/out-put volumes are also an important limitation in SEC. Ultracentrifugation and SEC are of limited accuracy, therefore they are commonly combined with other isolation methods including ultrafiltration [74].

### 3.3. Polymer Precipitation

This method has been routinely used for isolating viruses [75] and macromolecules for over 50 years. Polyethylene glycol (PEG) or other hydrophobic polymers precipitate exosomes through changing solubility and dispensability of the samples. Typically, PEG precipitation solution is combined with exosome containing biofluid and incubated at 4 °C overnight, with the obtained precipitation separated via low-speed centrifugation or filtration. Many companies offer isolation kits, such as ExoQuick or Pure-Exo. However, these methods co-precipitate contaminants, such as proteins and lipoproteins [76,77].

### 3.4. Immunoaffinity Chromatography

In this method, antibodies are attached to magnetic beads or other matrices through covalent bonding. The whole process depends on binding reaction between antibodies and specific surface-associated proteins expressed by exosomes, such as Alix, TSG101 or tetraspanins, resulting in immobilization of exosomes on magnetic beads. This method allows to isolate specific subpopulations of these nanovesicles, containing only antibody-recognised proteins, resulting in high purity isolate. Additionally, it can be applied for quantitative and qualitative analysis of exosomes. However, this is a relatively expensive method, abd ius not suitable for the isolation of large amounts of EVs [77,78].

### 3.5. Microfluidistics-Based Techniques

This method of separation is based on the physical and biochemical properties of particular exosome subtypes. Isolation techniques based on microfluidistics developed may be divided into three categories: immunoaffinity, sieving, and exosome separation using porous structures. Microfluid-based isolation techniques are in the early stages of development. However, due to their advantages, such as low reagent volumes, very high purity of isolated products and short processing time, they will be widely used in diagnostics. The main drawback for their clinical application is a problem with the fast and efficient production of sufficient exosome quantities. For exosome isolation and analysis, microfluidistics can further be combined with immunoaffinity and sieving [79]. 

### 3.6. Exosomes of Different Tissue and Cellular Origins

#### 3.6.1. Liver

Exosomes play an important role in communication between hepatocytes and non-parenchymal cells in liver tissue. Using a polymer precipitation technique, Nojima et al. isolated hepatocyte-derived exosomes containing sphingosine kinase 2 (SK2), a protein involved in liver repair and regeneration after injury. Exosomes derived from non-parenchymal cells did not induce those reactions, suggesting SK2 to be a specific cargo of hepatocyte-derived exosomes [80,81].

#### 3.6.2. Heart

Exosomes are considered to be involved in cardiac protection and repair [82,83]. Exosomes from cardiomyocytes were first isolated in 2007 from rat primary cell culture, using ultracentrifugation and differential centrifugation techniques [84]. Since then, a large number of studies described heart-derived exosomal cargo in physiological and pathological conditions [85]. In one of them, the researchers measured the level of four cardiac-specific miRNAs (miR-1, miR-208a, miR-133a, miR-499) in rat models of the cardiac fibrosis process and confirmed miR-208a participation in fibroblast proliferation and differentiation [86].

#### 3.6.3. Brain

Functions of exosomes in the central nervous system (CNS) are described in Section 3.2. The EV isolation protocol of exosomes present in the CNS extracellular space comprises the gentle dissociation of brain tissue, to avoid excessive cellular lysis, and application of low-speed centrifugation, filtration, and ultrafiltration, in that order [87,88]. Particular subpopulations of exosomes, containing unique cargo, are secreted by neural cells. Neuron-derived nanovesicles contain a specific protein, anti-neural cell adhesion molecule L1 (L1CAM) [89]. Oligodendrocytes-derived exosomes carry proteolipid protein (PLP), one of the major component of myelin [90]. Whereas microglia release vesicles enriched in CD13, surface-associated peptidase [91].

#### 3.6.4. Bone

Exosomes play a crucial role in bone homeostasis. Exosomes are secreted by almost every bone cell, such as bone mesenchymal stem cells, osteoblasts, osteoclasts, osteoclast precursor cells, osteocytes, bone marrow stromal cells and bone narrow adipocytes, thereby participating in osteogenesis, bone remodelling and resorption [92]. The isolation technique most commonly used to separate bone-derived exosomes is ultracentrifugation, whereas the analysis of bone-derived exosomal markers reveals specific molecules, such as osteoclast-derived miR-214-3p, reducing bone formation [93].

#### 3.6.5. Adipose Tissue

Exosome release has been described in adipocytes and adipose stromal cells (ADSCs), playing an important role in sustaining homeostasis, through participating in numerous processes: adipogenesis, angiogenesis, nerve regeneration, inflammation, regulation of energy metabolism and immunomodulation. Adipocyte-derived exosomes contain molecules such as adiponectin, resistin, tumour necrosis factor α (TNF-α), retinol binding protein 4 (RBP-4), macrophage-colony-stimulating factor (MCSF), fatty acid synthase (FASN), glucose-6-phosphate dehydrogenase (G6PD) and acetyl-CoA carboxylase (ACC) [94].

## 4. Clinical Significance of Exosomes

Recently, the potential application of exosomes as diagnostic targets has gained attention. How to take the advantages of exosomes in clinical applications is one of the important directions for exosome study. Their properties, such as the fact that exosome content changes significantly during morbid conditions in comparison to physiological state, has been used as a significant starting point in looking at exosomes as a potential diagnostic tool. Moreover, due to their common presence in biofluids, they are easily accessible in a non-invasive manner. Exosomes secreted by various cells express specific surface molecules, which can be used to determine their cellular origins. Additionally, these nanovesicles are stable and can be stored long-termm, as their content, used for analyses, is membrane-enclosed and protected from degradation [95]. For instance, loading a bioactive protein in exosomes avoids the immunogenicity and cleavage by proteases. Tang et al. recently engineered exosomal Tat to specifically reactivate latent human immunodeficiency virus 1 (HIV-1) [96]. Further, CD4+ exosomes released from CD4+ T cells seem to hinder spread of HIV virus by competing with viral particles in terms of binding to other molecules and therefore restrict viral replication by for example delivering active molecules [97]. Currently, it is clear that exosomes also are vehicles for HIV particles in acquired immune deficiency syndrome (AIDS) infected persons, contributing to the overall pathogenic effect. Even though the effect is still persistent, it is now possible to target the mechanisms involved in sorting of the viral particles into exosomes [98]. Alternatively, other studies have shown that exosomes carrying HIV-1 protein Nef (exNef) as their cargo are prone to be engulfed by macrophages, causing the release of the exNef into the cellular space [99]. 

### 4.1. Exosomes as a Diagnostic Tool-Diagnostics in CNS Diseases

In 2006, Faure et al. provided the first direct evidence of exosome secretion by neurons. Subsequently, numerous studies confirmed release of these nanovesicles by glial cells, such as astrocytes, oligodendrocytes, microglial cells, and neural stem cells. Exosomes are an important mode of communication between neurons and glia in the CNS, making them crucial for the physiological function of neurons through the participation in and modulation of numerous processes, including neuronal maturation and repair, as well as the activity and plasticity of synapses [100]. Exosome secretion by neurons and neuroglia is a reflection of their current state as they also readily cross the blood–brain barrier [101]. These properties mark exosomes as potential future diagnostic tool for neurodegenerative diseases. Currently, diagnosis of neurodegenerative diseases is based on clinical symptoms, leaving limited therapeutic possibilities due to the high advancement of the disease at the moment of detection. However, in 2006, Rajendran et al. described the presence of the Aβ protein (a toxic protein accumulated in brain parenchyma typical for Alzheimer’s disease (AD) in exosomes released by human embryonal carcinoma cell line NT2a, as well as the presence of an exosome-associated Alix protein in the amyloid plaque of three AD patients [102]. In 2011, an elevated level of phosphorylated tau protein (AT270) was found in neutrally derived exosomes extracted from the cerebrospinal fluid (CSF) of early AD patients [103]. In 2015, Goetzl et al. found out that increased level of cathepsin D, lysosomal-associated membrane protein 1 (LAMP1) and ubiquitinylated proteins, as well as lowered content of HSP70 in exosomes is related to pre-clinical phase of AD [104]. Multiple sclerosis, frontotemporal dementia, amyotrophic lateral sclerosis, Huntington’s disease and prion protein-associated diseases are all neurodegenerative illnesses in which exosomes and their content may serve as effective, pre-clinical diagnosis [105,106,107,108,109,110]. Acquired brain injuries could also be diagnosed based upon exosomes. Traumatic brain injury (TBI), spinal cord injury (SCI) and ischemic stroke lack specific biomarkers allowing for fast detection of their nature, extent and affected region. Exosomes carrying information on their cellular origins may provide these information, becoming potential novel biomarkers [111]. Exosomes are also a promising diagnostic tool in status epilepticus [112].

### 4.2. Exosomes and Cancer

Exosomes are considered to be involved in numerous mechanisms promoting cancer development, such as pre-metastasis niche formation [113], angiogenesis [114], migration and invasion [115], immune response modulation [116], metastasis [117], and drug resistance. Moreover, increased exosome secretion is a key adaptation to hypoxia, facilitating angiogenesis and metastasis in new niche conditions [118]. Small extracellular vesicle loading systems employing exosomes and exosome mimics known as small extracellular vesicles (sEVs) are being developed as a novel delivery strategy in chemotherapy-based cancer therapies dependent on loading external cargo composed of a tumour inhibiting agent and modifying exosomal surface proteins [119]. Recently designed artificial chimeric exosomes demonstrated a better antitumor therapeutic answer with elevated tumour accumulation when comparing with conventional liposomes [120]. Main challenges of exosome based systems include cancer-specific methods for loading the cargos into the vesicle and manipulation of the surface proteins so that the half-life of the vesicles is prolonged, making them a long lived therapeutic target to be explored in the near future. Additionally, recent research concerning the native content of the exosomal cargo might also significantly contribute to the understanding of exosomal proteomics, creating further possibilities for exploring alternative therapeutically relevant agents. As described, exosomes are deeply involved in cancer progression. Thus, the disruption of communication via tumour-derived exosomes is a potential therapeutic treatment strategy that can be achieved through the inhibition of exosome formation, release, or uptake by recipient cells. For example, the application of sphingomyelinase inhibitors, which participate in intracellular ceramide-dependent exosome synthesis, leads to decreased production of these nanoparticles [3]. The small GTPase family of RAB proteins is involved in exosome release and could be yet another potential group of treatment targets. RAB27A blockade was found to result in tumour growth inhibition [6]. Exosome secretion is also dependent on intracellular calcium levels, with its increase resulting in increased EV release. Application of dimethyl amiloride, an inhibitor of voltage-gated Ca2+ channels, results in myeloid-derived suppressor cell (MDSC) inhibition due to the decreased production of tumour-derived exosomes (TDEs), resulting in the reduced immunosuppressive function of these cells [121].

#### 4.2.1. Exosomes as Tumour Biomarkers

Solid biopsy is the most common tumour diagnosis tool used in clinical practice. However, this method is highly invasive, often unpleasant, and traumatic to a great number of patients. Additionally, biopsies are an impractical for conducting screening and prognostic assays. Hence, clinicians and diagnosticians have found better means of diagnosis, including the Food and Drug Administration (FDA) approved prostate intelliscore test, EV-based tests available for clinical use. Recently, far less invasive liquid biopsies are gaining more and more interest, with blood-derived or urine-derived exosomes indicated as novel potential diagnostic and prognostic markers for many types of cancer. Yoshioka et al. developed a new method for diagnosis of colorectal cancer, called “ExoScreen”. This method is highly sensitive, quick, and easy in terms of execution. Tumour-derived exosomes are trapped by two antibodies, one specific to CD9, a tetraspanin protein widely present on the exosome membrane, and one specific against CD147, a protein specific for colorectal-derived exosomes. Binding effectiveness can then be effectively detected with the use of immunoblotting [122]. Another interesting study identifies glypican-1 (GPC1) as a biomarker of early state pancreatic cancer [123]. Several types of exosomes and their cargo (especially miRNA) are also used as preclinical biomarkers in many types of cancer, such as lung cancer, hepatocellular carcinoma, pancreatic cancer, colorectal cancer, melanoma, breast cancer, prostate cancer, ovarian cancer, glioblastoma, and nasopharyngeal carcinoma [72,124].

#### 4.2.2. Exosomes as a Drug Delivery System

Currently, drugs and genes are most commonly administered with the use of liposomes and polymeric nanoparticles. Liposomes are synthetic, sphere-shaped phospholipid nanovesicles composed of at least one lipid bilayer that encloses an aqueous space [125]. Nanoparticles are synthetic or semi-synthetic colloidal polymers of 10–1000 nm in diameter [126]. Besides the fact that these vesicles have been commonly used as a promising administration route for many anti-cancer drugs, anti-fungal drugs, and analgesics, limitations in their usage include the short half-life of liposomes in the circulatory system, varying biocompatibility, and long-term toxicity. Exosomes seem to be a potent drug delivery system with an array of desirable features being composed of membranes rather than synthetic polymers, which represent improved tolerance by host organism. Exosomal vesicles have promising long circulating half-life, very low or no toxicity, intrinsic capability to target specific tissues or even cells (very important in CNS diseases), low immunogenicity and tend to have innate homing capacity [127]. Exosomes may also be genetically engineered to pass through biological obstacles, such as the blood-brain barrier, penetrate into tissues, as well as carry numerous types of drug molecules and genes, such as proteins, lipids, RNAs and DNAs, effectively protecting them from degradation [128]. One of the first studies describing the usage of exosomes as a drug delivery system was conducted in 2010. Curcumin, an anti-inflammatory, antineoplastic and antioxidant drug, was administered to mice with lipopolysaccharide-induced septic shock. It was found that curcumin carried by exosomes was more soluble and bioavailable with higher clinical activity [129]. Since then, numerous studies on animal models and cell cultures have been performed, such as: exosome mediated catalase administration to in vitro cultured neurons and mice suffering from PD [130], doxorubicin administration to human H1299 and A549 lung cancer cells [131], or delivery of miRNA (miR-122) to human liver hepatocellular carcinoma HepG2 cell line, as well as HepG2 cell bearing mice in vivo [132]. Due to the success of these studies, exosomes have recently been approved as drug carriers in clinical trials, facilitating the treatment of melanoma, non-small-cell lung carcinoma, colorectal cancer, cancer of the head and neck, as well as ulcers and type I diabetes mellitus [70,72]. Critical challenges in the context of exosomes as potential drug delivery vehicles for cancer therapies include ineffective exosome separation techniques and a lack of purification techniques required following the successful isolation of exosomes [119]. Other challenges encompass the limited availability of highly sensitive exosomal biomarkers, non-large-scale production and low drug loading efficiency [120]. 

## 5. Conclusions

In summary, exosomes are the smallest group of nanovesicles, serving as a method of intracellular communication. Interest in exosomes has grown tremendously in the past two decades due to their multiple functions in both physiological and pathological processes. Exosomes not only allow cells to send out signals, but also to transport proteins. Exosomes can either be formed in the ESCRT dependent or ESCRT—independent pathways. Currently, the most effective methods for the isolation of exosomes of statistically relevant purity include ultracentrifugation, size exclusion chromatography, polyethylene glycol precipitation, immunoaffinity chromatography, and microfluidistics. Exosomal cargo vary according to the organ they are derived from. There is a rapidly growing interest in employing exosomes as diagnostic tools for cancer and tumour therapies as well as vectors for drug delivery. There are still more questions than answers in the subject of their biogenesis, cargo sorting and release mechanisms. Moreover, the isolation of clinical grade exosomes is time-consuming, requiring the development of new, more efficient technologies. The understanding of exosome biology is improving, creating a new area for diagnostic, prognostic, and therapeutic applications to be discovered in the near future. 

## Figures and Tables

**Figure 1 jcm-09-00436-f001:**
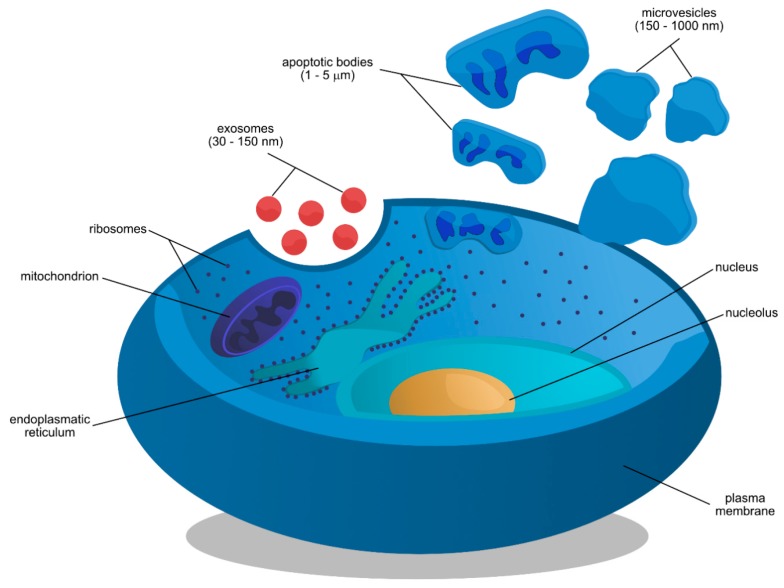
Biogenesis of three types of extracellular vesicles including exosomes, apoptotic bodies and microvesicles. All nanovesicles are released into the extracellular space, however their synthesis is dependent on the state of cell, e.g., apoptotic bodies are only produced during programmed cell death, while exosomes and microvesicles are secreted during cell cycle and normal state of cell.

**Figure 2 jcm-09-00436-f002:**
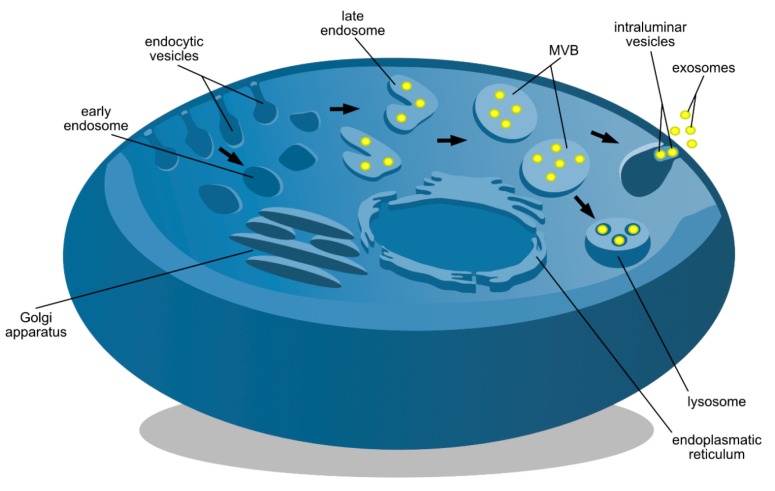
Endocytic pathways in the process of exosome biogenesis. Different steps of exosome biogenesis are demonstrated including early endosome formation, late endosome formation and multivesicular body (MVB) formation, respectively. MVB is then either transported into the lysosome for lysosomal exocytosis or fused with the endosomal membrane followed by exosomes release into the extracellular space. Two main organelles are shown including Golgi apparatus and endoplasmic reticulum due to their interaction with early endosomes as soon as they are formed from endocytic vesicles.

**Figure 3 jcm-09-00436-f003:**
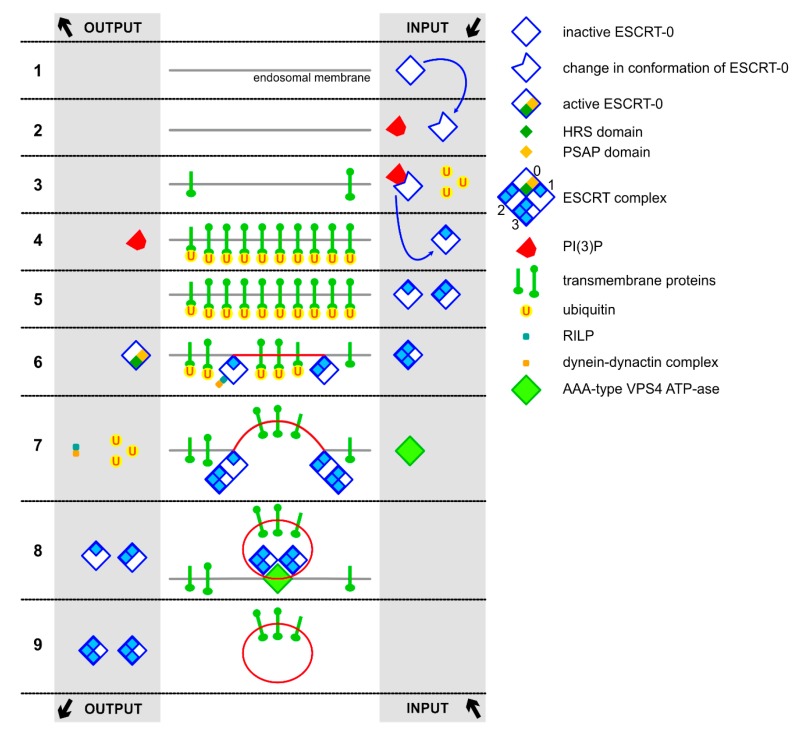
**The endosomal sorting complexes required for transport** (ESCRT)-dependent mechanism of sorting of ubiquitinated cargo into the multivesicular precursor of exosomes. The process involves accumulation and ubiquitination of transmembrane proteins on to the late endosomal membrane, followed by recruitment of phosphatidylinositol 3-phosphate (PI(3)P), ESCRT complex (ESCRT-0, ESCRT-1, ESCRT0-2, ESCRT-3), RAB-interacting lysosomal protein (RILP), dynein-dynactin complex, AAA-type vacuolar protein sorting-associated protein 4 (VPS4) ATP-ase, respectively. Following steps of the ESCRT-dependent sorting are presented including the input, output and changes in conformation of particular proteins due to their activation, stimulation and reuse. Importantly, upon change in conformation of ESCRT-0 PI(3)P protein binds and allows ESCRT-0 dependent activation of ESCRT-1. Active ESCRT-1 signals back to ESCRT-2 that works in association with RILP protein-dynein-dynactin complex. Once both ESCRT-1 and ESCRT-2 are localized on to the late endosomal membrane, ESCRT-3 is recruited at both sites to facilitate pinching in the membrane. AAA-type VPS4 ATP-ase comes at last to pinch off the endosomal vesicle and release it in association with ubiquitinated cargo.

**Figure 4 jcm-09-00436-f004:**
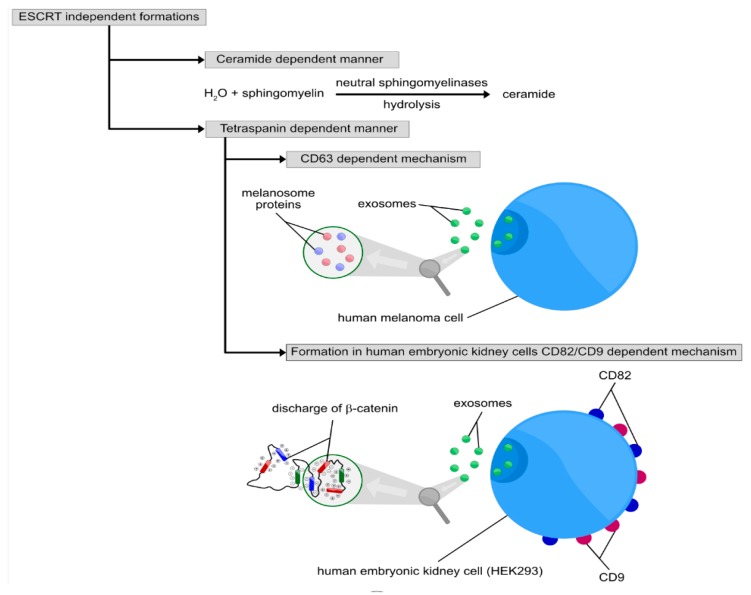
Three independent of each other processes of ESCRT-independent formation are proferred, including ceramide dependent manner, cluster of differentiation 63 (CD-63) dependent mechanism and ESCRT-independent formation in human embryonic kidney cells (HEK293). The ceramide dependent manner of ESCRT-independent formation relies on hydrolysis of spingomyelin to ceramide associated by the presence of neutral sphingomyelinases (nsMase). CD-63 dependent mechanism of ESCRT-independent mechanism is associated with release of exosomal cargo containing melanosome proteins from the exosomes secreted by human melanoma cells. ESCRT independent exosomal nanovesicle formation in HEK293 relies on discharge of β-catenin through the membrane of exosomes secreted by CD82+ CD9+ HEK293 cell.

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
