# Peer review of "Inclusion Biogenesis, Methods of Isolation and Clinical Application of Human Cellular Exosomes"

_jcm, 2020, doi:10.3390/jcm9020436_

Round 1

Reviewer 1 Report

In the manuscript entitled " Biochemical properties, methods of isolation and clinical application of human cellular exosomes", Max Tschuschke et al introduced the biogenesis, properties, isolation of human exosomes and their clinical applications. Exosomes are naturally occurring nanoparticles in human bodies that carry numerous biological molecules to communicate among different cells. How to take the advantages of exosomes in clinical applications is one of the important directions for exosome study. Loading a bioactive protein in exosomes can avoid the immunogenicity and cleavage by proteases.For example, Ramratnam lab recently engineered exosomal Tat to specifically reactivate latent HIV-1(JCI Insight 2018 Apr 5;3(7).pii:95676.).  The authors should introduce more about the applications of exosomes in some important human diseases such as HIV/AIDS and cancers.  

Author Response

Dear Reviewer,

We would like to thank you for your comments and suggestion.

According to your advices, the mentioned issue was incorporated in the body of manuscript (361-371). 

We hope you will be satisfied and find the manuscript suitable for publication.

Best regards,

Ievgeniia Kocherova

Reviewer 2 Report

The authors describe the exosome biogenesis, their cargo, and methods of isolation; furthermore, the clinical relevance of exosomes has been reported in the concluding part of the paper, but they have hinted at the most exciting questions only, lack an accurate conclusion.

In the abstract and introduction, the concepts are not ever reported with a logical sense; for example, in the initial presentation of EVs concerning biogenesis, the authors begin to illustrate the microvesicles then the apoptotic bodies and in the end exosomes, without following any criterions, neither dimensional parameter. Moreover, the authors claimed that: Main division is based on the process of biogenesis and biophysical features, like density and membrane composition [3]. However, there is not information about biophysical features, like density and membrane composition. Moreover, the reference: Lee, Y.; Andaloussi, S.E.L.; Wood, M.J.A. Exosomes and microvesicles: extracellular vesicles for genetic information transfer and gene therapy. 2012, 21, 125–134, does not seem to correspond to what has been stated fully; there are more recent and relevant articles that should be mentioned. The other used references are not recent also, and I am not referring to those relating to the initial data of their study.

In the paragraph: “Cellular origins and chemical properties of exosomes”, the mechanisms of biogenesis are described unclearly, and old references are reported, while there are many updated articles about this issue

Page 3, line 91, the sentence: “Most of the time, MVBs are directed to hydrolase containing lysosomes, resulting in degradation of their cargo” is confounding, in fact the hydrolase are contained within the lysosomes and not the opposite.

In the figure 2 the lysosomal compartment isn’t reported, then the scheme is incomplete

The Authors describe the mechanisms involved in the formation of exosomes reporting ESCRT-dependent and –independent systems; however, the descriptions are incredibly synthetic, risking being unclear. Moreover, some recent articles and reviews are not mentioned; they could be useful for those wishing to deepen the argument. For example: Cell Mol Life Sci. 2018 Jan;75(2):193-208. doi: 10.1007/s00018-017-2595-9.Current knowledge on exosome biogenesis and release. Hessvik NP, Llorente A.

The paragraph “Cargo” doesn’t report data from recent articles, the authors should update the bibliography. Moreover, the associations between cargo composition and clinical relevance are not reported.

The role of exosomes in the pathogenesis of some diseases such as cancer isn’t reported and no reference is illustrated on the link between exosomes and autophagy which is interconnected significantly to endolysosomal pathway.

In general, the authors seem show more interest to reporting previous data by not giving space to the most updated literature results

Minor revisions:

Page 1, Line 35: In the sentence: Endosomal Sorting Complex Required for Transport (ESCRT) and ECRT-independent system, the acronym ECRT is wrong, change with ESCRT

Page 2, Line 78: lipid change with lipids

Page 5, line 137: “an ESCRT-independent pathways”

Page 6 line: CD-63 uniform without dash

Author Response

Dear Reviewer,

We would like to thank you for your valuable comments to our manuscript.

According to your advices, the mentioned issues were clarified and incorporated in the body of manuscript.

We hope that improvement of manuscript is appropriate and increases scientific value of this review.

We hope you will be satisfied and find the manuscript suitable for publication.

Best regards,

Ievgeniia Kocherova

Reviewer 3 Report

The review ”Biochemical properties, methods of isolation and clinical application of human cellular exosomes” by Tschuschke et al addresses the current knowledge on extracellular vesicle biogenesis and their potential for clinical applications. The topic is important and broad and the text is relatively fluent. However, multiple reviews of the topics already exist and therefore the text should be improved by addition of references, clarification of the text in some parts and expression of the current limitations in addition to potential of the field. I have several specific comments:

-consider modifying the title, inclusion of biogenesis? “Biochemical properties” bring aspects such as charge, solubility etc in mind that are not addressed in the text.

-introduction:

-r 49-51 complicated sentence, clarify.

-EV sizes; it should be expressed that the size ranges are rough estimates.

-Include/introduce/mention exomeres as a novel EV subclass? Relates to Fig.1 as well.

-Exocarta molecular data mentioned – do other databases (Vesiclepedia, EVpedia) have more or less entries?

-2.2:

-r.142-144: clarify sentence (neither… nor…)

-ESCRT independent release mechanisms: it seems a little unclear, how the pathways should be called – in fig.4, formation in HEK cells hardly is a pathway name (the type of secretion probably is not only found in HEK cells?). Fig.4 text starts also with complicated sentences related to the trouble of naming the pathways. It should be also mentioned in the first sentence that the fig. deals with formation of EVs. Related to that, could the CD63 and CD9/CD82 pathways be called tetraspanin-dependent routes?

-2.3:

-there are many unclear points, r. 172-176 sentence first expresses that post-translational ISGylation of TSG101 leads to impaired exosome release, but then also to increased release?

-r.176: clarify “centrifugal transport”

-r. 184: clarify “release of different exosomes requires various RAB proteins”, what is meant by different here?

-r. 186-187, clarify “facilitate fusion of both vesicles together…”, what is meant by both vesicles?

-2.4:

-cup-shape of EVs is observed after negative staining or other non-cryo EM processing, i.e. it has been interpreted as an artefact of this processing as the cup-shape is not visible by cryo-EM

-add ref to proteins crucial for MVB biogenesis (e.g. clathrin)

-rephrase “lipids are the most commonly found molecules”? Why do authors say so? The EVs always have a lipid encapsulation, but lipidomics is not the most often used method in EV analysis.

-add discussion and ref at least to DNA part, e.g. Jeppesen et al., 2019, where DNA was found not to be incorporated in exosomes, perhaps add discussion, where the molecules may be found (in or outside of EVs in the EV prep)

3.1:

-list of fluids where exosomes have been isolated is not complete, thus add “for example”

-clarify that the isolation methods do not only isolate exosomes or nanovesicles

3.1.1:

-three centrifugation steps is an example that can be used, not a gold standard

-second centrifugation step 10-20 000g also pellets MVs

-gradient medium is not (only) mixed with the samples, clarify this separation method

3.1.2:

-In/out-put volumes are also an important limitation in SEC

3.1.3:

-PEG precipitation is hardly very expensive, I would recommend to remove this limitation.

3.1.4:

-relatively expensive method and not suitable for isolation of large amounts of EVs, these could be mentioned here

3.1.5

-microfluidistics

-is not divided, but can be combined with immunoaffinity, sieving etc, this should be clarified, add also reference here

3.2:

-what is meant by specificity here? Presence of some cargo in exosomes from one cell line but not in exosomes of another cell line, does not yet mean specific.

4.1

-add references

4.1.1:

-add ref to the ability of exosomes to cross blood-brain barrier

-Hela is not brain derived, so what is the point to mention that Abeta protein is found in the Hela EVs?

4.2

-add references to the role of exosomes in drug resistance or higher secretion in hypoxia

4.2.2

-text lacks the FDA approved prostate intelliscore test and it would be good to add as there are not yet many EV-based tests available for clinical use

4.2.3

-text lacks the acknowledgement of problems/challenges in drug delivery by exosomes and mentions all the potential/promise as facts (targeting and homing capacity…etc), the text should be changed to give a realistic view of the current status of this field

-General:

-complicated sentence and spell check required, abbreviations and some typos, such as bone narrow to bone marrow, microfluids to microfluidistics

Author Response

Dear Reviewer,

We would like to thank you for your valuable comments to our manuscript.

According to your advices, the mentioned issues were clarified and incorporated in the body of manuscript. The new references have been added.

We hope that improvement of manuscript is appropriate and increases scientific value of this review.

We hope you will be satisfied and find the manuscript suitable for publication.

Best regards,

Ievgeniia Kocherova

Round 2

Reviewer 2 Report

The authors did not respond adequately to my criticisms; several critical issues persist. In my pure opinion, the present work is not adequate for publication